# OpenReview forum: "Can large language models explore in-context?"
_ICML.cc/2024/Workshop/ICL — ICML 2024 Workshop ICL Poster_

### Official Review · Reviewer_c6uC · 2024-06-08
**Prompt engineering in an interesting reinforcement learning setup**

**Rating:** 2
**Fit:** 3
**Confidence:** 2

**Workshop Review:**

The work investigates LLM's exploration abilities, an important aspect in the reinforcement learning world. The formulation is similar to classification task with unseen/OOD labels, and happens across certain time horizon. The setup of using LLM for exploration is interesting. However, the methods are mostly prompt engineering and lack some novelty. For the clarity, I would recommend the authors to follow conventional ways of dividing sections, e.g. to have a conclusion section, to better organize the info.

**Reason For Not Giving Higher Score:**

- The methods of prompt engineering lacks novelty.

**Reason For Not Giving Lower Score:**

- The topic of using LLM for exploration is interesting.

---

### Official Review · Reviewer_SN3H · 2024-06-10
**Good work on in-context RL with pre-trained LLMs**

**Rating:** 3
**Fit:** 3
**Confidence:** 2

**Workshop Review:**

This work studies an important question, whether LLMs can explore in-context. It goes beyond prior work by characterizing distinct failure modes - suffix failures and uniform - and comparing LLM behavior to classic RL models such as Thompson sampling and UCB. This setup paired with ablation experiments across 5 prompt configuration options provides novel insight into how and why failures happen. The results are interesting and the writing describes them well. Figures 1 and 3 were effective in conveying results.

Strengths:
1. Experiments are fairly thorough, testing a large number of prompting and task variations
2. Good setup for ICRL, using pre-trained LLMs with no fine tuning for multi-armed bandit tasks
3. In-depth study of failure modes and comparison to ground truth models provides deeper insights into ICRL behavior

Weaknesses:
1. This doesn't cite [1, 2] which also study multi (two) -armed bandit tasks in pre-trained LLMs with prompting only. [1] has a similar finding, that GPT-3 fails to explore. From what I can tell, this work goes beyond [1] in a number of ways: >2 armed bandits, the study of failure modes, comparison across prompt configurations, and comparison to baselines such as UCB and TS.
2. There isn't much discussion of how these results relate to LLM understanding in general, which might help ground the impact of the findings. What do suffix or uniform failures mean for LLM agent design, or what would they look like in a typical LLM use case?
3. It might help to move some of the figures from the appendix, such as Figure 10, to the main text. Some text such as "Scale of the experiments." could be shortened or moved to the appendix.



[1] Using cognitive psychology to understand GPT-3 (2023)

[2] Meta-in-context learning in large language models (2023)

**Reason For Not Giving Higher Score:**

See weaknesses.

**Reason For Not Giving Lower Score:**

See strengths.

---

### Meta-Review · Area_Chair_STjz · 2024-06-14

**Recommendation:** 2

**Metareview:**

This paper investigates the exploratory capabilities of Large Language Models (LLMs) within multi-armed bandit environments, without using training i.e. in-context reinforcement learning. The authors come to a mixed conclusion finding both positive and negative evidence of the exploration capabilities in LLMs.

Both reviewers find the setup for analysis of ICRL interesting and reviewer SN3H points out the thoroughness of the experiments. SN3H points out missing relevant citations and suggests that the discussion could benefit from a broader contextualization of the results in terms of general LLM understanding. c6uC finds the non-standard formatting of the paper confusing which should be addressed.

Overall, the paper meets the threshold for inclusion in the workshop.

---

### Decision · Program_Chairs · 2024-06-17

Accept (Poster)